# Urdbean Leaf Crinkle Virus: A Mystery Waiting to Be Solved

**DOI:** 10.3390/v15102120

**Published:** 2023-10-19

**Authors:** Naimuddin Kamaal, Mohammad Akram, Aditya Pratap, Deepender Kumar, Ramakrishnan M. Nair

**Affiliations:** 1ICAR-Indian Institute of Pulses Research, Kanpur 208024, India; naimuddin@icar.gov.in (N.K.); aditya.pratap@icar.gov.in (A.P.); deependerkr@gmail.com (D.K.); 2World Vegetable Center, South Asia, ICRISAT Campus, Hyderabad 502324, India; ramakrishnan.nair@worldveg.org

**Keywords:** virus, blackgram, *Vigna mungo*, insect vectors, leaf crinkle

## Abstract

Urdbean leaf crinkle disease (ULCD) affects mainly the urdbean or blackgram (*Vigna mungo* (L.) Hepper) causing distinct symptoms that often result in serious yield losses. It has been known to occur for more than five decades and is considered to be of viral etiology. The identity of the causal agent, often referred to as the urdbean leaf crinkle virus, is not unequivocally proved. There are few attempts to characterize the causal agent of ULCD; however, there is no unanimity in the results. Recent attempts to characterize the causal agent of ULCD using next-generation sequencing of the virome of ULCD-affected urdbean plants indicated the involvement of cowpea mild mottle virus; however, without conforming through Koch’s postulates, the etiology of ULCD remains inconclusive. Claims of different insect vectors involved in the transmission of ULCD make this disease even more mysterious. The information available so far indicates that either two different viruses are causing ULCD or a mixture of viruses is involved. The identity of the virus/es causing ULCD still remains to be unambiguously ascertained. In this review, we attempt to analyze information on the various aspects of ULCD.

## 1. Introduction

Urdbean (*Vigna mungo* (L.) Hepper, also known as blackgram, is an important pulse crop of the Indian sub-continent. In India, it is mainly grown during the kharif (rainy) season in the northern and central parts. Additionally, it is also grown after paddies as a rice fallow crop in large tracts of the southern states during the rabi (winter) season and is also a catch crop during the spring season in the northern plains. The urdbean was cultivated in over 4.63 m of hectares, with a total production of 2.78 million tonnes and a productivity of 600 kg/hectare in India in 2020–2021 [1], contributing around 10% to the total pulse basket of the country. Besides India, the urdbean is also cultivated in Myanmar, Bangladesh, China, Pakistan, Thailand and Sri Lanka [2]. In Myanmar, it is mainly grown in the lower regions of the Bago and Ayeyarwaddy areas [3], while Pakistan has also recently emerged as a large exporter of the urdbean [2]. The urdbean is rich in protein and carbohydrates and consumed in a variety of ways. This crop has a yield potential of more than 1.5 t but its average productivity hovers around 0.6 t. Of the various abiotic and biotic factors which inflict crop at various stages, diseases caused by viruses are the major biotic constraints faced by this crop. Among virus diseases, while yellow mosaic disease caused by many begomoviruses [4] is considered as the most important disease of the urdbean, the leaf crinkle disease that is widely prevalent in India and Pakistan is capable of causing crop losses up to 100% depending upon the season and the variety [5,6]. It may emerge as the most serious disease because of its debilitating effect on the plant and being seed borne. ULCD was first reported in Delhi in 1966 by Williams et al. (1968) [7] and in the Tarai region of the then Uttar Pradesh in 1967 [8]. Subsequent investigators [9,10] concluded ULCD to be of a virus etiology and named the causal virus as the urdbean leaf crinkle virus (ULCV); it has not been classified taxonomically [11] and whether ULCV is a distinct virus is yet to be established. Of late, there have been a few attempts to crack the identity of the causal virus or viruses of ULCD, but without any conclusive results. Since there is no record of a virus named ULCV by International Committee on Taxonomy of Viruses (ICTV) [12], designating the causal agent of ULCD as ULCV does not appear logical and justified. In the present review, therefore, the use of ‘causal agent of ULCD’ or simply ULCD has been preferred over ULCV. The current status of understanding on various aspects of this mysterious disease is reviewed here and the schematic presentation of the above aspects is presented in Figure 1. 

## 2. Disease Symptoms

Characteristic symptoms of ULCD include unusual enlargement of the leaves, often described as crinkling, curling or rugosity [7,8,28]. The disease affects both vegetative as well as reproductive parts, often rendering the plant completely unproductive. The first visible symptom under field conditions is generally observed in the third trifoliate which appears light green with slight enlargement of the leaflets and raising of inter-venal areas. Subsequently emerging leaves are malformed and develop typical crinkle symptoms, i.e., an enlargement in size and crinkled surface of lamina [14]. Enlargement of the leaf lamina may be one and half time larger than the healthy ones [21]. The shortening of internodes often leads to clumping of apical growth. Flowering in a crinkled plant is delayed and inflorescence looks bushy due to clumping (Figure 2). It is not uncommon to observe a malformation of floral parts. Reduced pod setting in the crinkle-affected plants has been attributed to ULCD’s adverse effect on pollen fertility [21,29]. Leaf crinkling, witches’ broom symptoms and sterility [30], and malformation of inflorescence with proliferation of flowers with complete sterility [10,31] are also reported. In the field, patches of crinkle-affected plants are often noticed. ULCD-affected plants produced light brown-colored seed as compared to the normal black color and were shriveled [32]. ULCD causes a reduction in number, weight and size of root nodules in the urdbean [31]. Several studies concluded that ULCD symptom expression is heavily influenced by temperature, resulting in corresponding yield reductions [33,34,35,36,37,38,39,40]. Studies on the effect of ULCD on the nutrient content of the affected urdbean plant indicated increased phosphorus and potassium and decreased nitrogen, calcium and magnesium in leaves [41]. Increased level of Indole-3-acetic acid (IAA) in ULCD-affected urdbean leaves has also been reported [42]. 

## 3. Yield Losses Due to ULCD

Urdbean leaf crinkle disease incidence is reported to vary from season to season [43,44] and yield loss thus depends on the season and the genotype [32]. Yield loss also depends on the age at which the plant becomes infected. Plants infected at early stages suffer more damage than those infected at later stages of growth [32,45]. This is probably the reason that different levels of yield losses are reported by different workers [20,37,43,44,46,47]. It is nevertheless well accepted that the plant of a susceptible variety infected at an early stage of growth suffers more loss, often becoming unproductive. The reduction in yield is attributed to the reduced number of pods on the crinkle-affected urdbean plant [46,47]. Beniwal and Chaube (1979) [32] and Singh (1980) [48] reported 76–100% yield loss due to crinkle in the urdbean, whereas in Pakistan, Bashir et al. (1991) [49] reported 35–81% reduction in seed yield and added that the loss depended on the genotype and age of the plant at infection. Kadian (1982) [45] found that the extent of yield loss depended on the growth stage at which the plant was infected. Grain yield loss of 2.12–93.98% in the mungbean cv. Varsha and 2.82–95.17% in the urdbean cv. T9 was recorded and was mainly attributed to the reduction in the number of pods in crinkle-affected plants and the stage at which the plant was infected. 

## 4. Epidemiology of ULCD

The effect of different environmental factors on the development of ULCD has been studied by few researchers. Kadian (1989) [50], working at Hisar, Haryana in India, found that ULCD in the urdbean became serious during kharif season and did not appear at all in the summer season or if it did appear, symptoms were very mild. The maximum disease incidence was observed during kharif season when the maximum and minimum temperatures were around 35 °C ± 2 °C and 25 °C ± 2 °C, respectively, and with a morning and evening relative humidity (RH) of >70% and >50%, respectively. During summers, a masking of crinkle symptoms was noticed when temperatures ranged between 38–45 °C along with a morning and evening RH of about 60% and 40%, respectively. Another study on the impact of environmental factors on ULCD development revealed a significant correlation with maximum and minimum temperatures and no correlation with weekly relative humidity, rainfall and wind movement. Maximum temperatures of 35–42 °C and minimum temperatures of 21–29 °C were conducive for disease epidemics during spring and summer sown crops [51]. Under glasshouse conditions, symptoms of ULCD in mechanically inoculated plants of the urdbean developed in a temperature range of 25–38 °C, with disease incidence being significantly higher between temperature ranges of 30–35 °C, which became reduced drastically at temperatures over 38 °C. The effect of temperature on ULCD was fitted to a non-linear beta model which can be used for predicting potential geographical areas for low disease incidence which in turn can be exploited for disease management under changing climates [34]. Managing the occurrence of ULCD requires a comprehensive understanding of climate conditions, weather parameters, host vulnerability and the development of disease forecasting systems. By considering these environmental factors, effective strategies can be implemented to mitigate the impact of ULCD on urdbean and mungbean crops.

## 5. ULCD Transmission

A review of early publications on ULCD indicates contradictions in regards to mechanical and vector transmission. Presently, the mechanical and seed transmission of ULCD is unequivocally proved but there is no unanimity on the vector.

### 5.1. Mechanical Sap Inoculation

Initially, attempts to transmit ULCD through mechanical sap inoculations were met with negative results [7]. It was, however, soon established that the causal agent of ULCD is mechanically sap transmissible [22,52,53,54], and many workers also reported its transmission by mechanical sap inoculation in later studies [6,18,55,56]. Using mechanical sap inoculations, it has been shown that the susceptibility of urdbean plants to leaf crinkle disease decreases with the increase in plant age and the incubation period increases with the increase in plant age at the time of inoculation [57]. Chowdhury and Nath (1983) [31] developed a method of inoculation of the ULCD causal agent, wherein germinated seeds were soaked and shaken in sap from ULCD-affected urdbean leaves. A lower disease incidence with reduced severity was observed for inoculation at a later stage. Interestingly, 100% disease incidence with very high severity was observed when inoculation was conducted in germinated seed [18]. It was also found that the drenching of urdbean leaves with 1% of Benlate resulted in reduced aphid and mechanical sap inoculation [58]. 

### 5.2. Seed Transmission

Seed transmission of ULCD has been proven conclusively [22,35,36,38,53,54,55,57]. The plant susceptibility and seed transmission (%) of the ULCD causal agent was reduced as the age of plants increased, indicating a negative linear relationship with age of the plant at the time of inoculation [29]. ULCD is also known to affect mungbean, and its causal agent has also been found to be seed borne (6–15%) in 3 of the 49 germplasm lines of the mungbean [13]. The age of the plant at the time of infection influences the rate of seed transmission. Plants infected at an early stage showed a higher rate of seed transmission than those infected in a later growth stage [59]. The percentage of seed transmission in urdbean cultivars viz., Azad Urd-2, LBG 623, T 9 and Uttara was observed in the range of 1.1–28.6% [6]. A recent publication reports ULCD to be transmitted through infected seed to the next generation at a rate from 40 to 54 percent without affecting seed germination [24].

### 5.3. Vector Transmission

Transmission of ULCD by the insect vector has been the most baffling issue, as different insects have been shown to be the vector of ULCD. It was shown to be transmitted by whitefly (*Bemisia tabaci*), which transmits the ULCD without any incubation period. An incubation period of 20–23 days is, however, required for symptom appearance [60]. Prasad et al. (1998) [25] also reported the whitefly transmission of ULCD from Meghalaya. On the contrary, *Aphis craccivora* Koch, *A. gossypii* Glov., *Acyrthosiphon pisum* and *Myzus persicae* [17,61,62,63,64] are also reported to be the vectors of ULCD. A pre-acquisition fasting increased the transmission efficiency of aphids which required 10-20 min of acquisition feeding and a short inoculation feeding period of 1–2 min [61,63]. 

Issue of vector transmission became further confounded when the Pantnagar isolate of the causal agent of ULCD was shown to be transmitted by a beetle, *Henosepilachna dodecastigma*, but not by aphids (*Aphis craccivora* and *A. gossypii*), whiteflies (*Bemisia tabaci*) and leaf hoppers (*Circulifer tennellus*) [65]. Adding more confusion to the differences on the issue of vector of ULCD, Brar and Rataul (1987) [15,16], based on their field and laboratory studies, concluded that ULCD is not transmitted by insects such as aphids (*Aphis craccivora*, *A. gossypii*, *Rhopalosiphum maidis*), whiteflies (*Bemisia tabaci*), jassids (*Empoasca motti*) and beetles (*Henosepilachna dodecastigma*). Nevertheless, reports of ULCD being transmitted by different vectors point to the possibility of involvement of different viruses in causing ULCD. A comparatively recent study [40] on the transmission of ULCD by different insects viz., aphids, leafhopper, whiteflies and beetles revealed that the causal agent of ULCD was effectively transmitted by one aphid species (*Aphis craccivora*) in a non-persistent manner and also by whiteflies (*Bemisia tabaci*). Other species of aphids, melon aphids (*Aphis gossypii*), cabbage aphids (*Brevicoryne brassicae*) and hadda beetles (*Epilachna vignitioctopuntata* and *Epilacna dodecastigma*) failed to transmit ULCD. On the other hand, Golluru et al. (2017) [64] showed that the aphid, *Aphis craccivora*, transmitted the causal agent of ULCD in non-persistent manner, unlike the whitefly, *Bemisia tabaci*. 

## 6. Host Range

Many workers have worked out the host range of the causal agent of ULCD. Using mechanical sap inoculation, Kolte and Nene (1975) [23] inoculated 52 plant species and found that only three (*Vigna radiata*, *V. sinensis* and *V. aconitifolia*) developed crinkle symptoms. In another study involving 60 plant species of 9 families, two cucurbitaceous plants viz., the cucumber and bottle gourd, were found to be the host of the causal agent of ULCD [59]. Kadian (1983) [19] found *Convolvulus arvensis* as a host with systemic symptoms and four species of *Datura viz*., *D. metel*, *D. stramonium*, *D. metaloides*, and *D. incrimis* as susceptible hosts with local symptoms while studying the host range of ULCD in most commonly found weeds., Seventeen weed species, namely *Amaranthus* spp., *Achyranthes* spp., *Cannabis sativa* (L), punarnava (*Boerhavia diffusa*), tree spinach (*Chenopodium giganteum*), quinoa (*Chenopodium quinoa*), silver cock’s comb (*Celosia argentea*), nut grass (*Cyperus rotundus*), asthma-plant (*Euphorbia hitra*), Philippine tea tree (*Euphorbia microphylla*), Forsk (*Digera arvensis*), tex-Mex tobacco (*Nicotiana plumbaginifolia*), gale of the wind (*Phyllanthus niruri*), common Purslane (*Portulaca oleracea*), Indian milkwort (*Polygala chinensis*), giant pigweed (*Trianthema monogyna*) and rough cocklebur (*Xanthium strumarium*) did not exhibit any symptoms [19].

## 7. Resistance and Physiochemical Changes in Host Plant

Several researchers have reported the reaction of mungbean and urdbean genotypes against ULCD under field conditions. Sravika et al. (2019) [66] found that the leaf crinkle disease in mungbean was more pronounced in rabi season at Coimbatore and found that genotypes RME-16-3, RME-16-12, MLT-GG R-16-007, MLT-GG R-16-009 and COGG 1319 (rabi-greengram) were highly resistant, whereas none of the greengram genotypes showed a highly resistant reaction in the *kharif* season. The mixed infection of crinkle with yellow mosaic disease and leaf curl (GBNV) in the urdbean also occurs in certain field conditions [67]. Godara et al. (2014) [68] screened mungbean genotypes and found none of them to be resistant. Seven genotypes of the urdbean viz., P-102, P-104, P-105, P-108, P-109, PU 14-1 and PU14-7, showed resistance against leaf crinkle and could be exploited in a resistance breeding program [69]. Kumar et al. (2013) [70] screened urdbean genotypes at Pantnagar and found the genotypes Sh. U-9503, Sh U-9518, PLU—289, PU19, PU-35 and NU-1 to be immune to leaf crinkle disease. Urdbean genotypes, PDKV Black Gold, Phule U-0609-43 and TBU-4 exhibited a resistant reaction under field conditions at Rahuri, Maharashtra [24]. Chaudhry et al. (2007) [71] screened 67 genotypes of the urdbean in Pakistan and found none of them to be resistant to ULCD. They, however, found two genotypes, i.e., 3CM-707 and CH-Mash 97, to be moderately resistant. Ashfaq et al. (2007) [72] screened 87 lines of the urdbean against ULCD and identified nine genotypes viz., 2 cm-703, 90 cm-015, 93 cm-006, 94 cm-019, 99 cm-001, IAM 382-1, IAM382-9, IAM382-15, and IAM133, as highly resistant, with 19 others as resistant. Bashir et al. (2005) [73], in Pakistan, found five mungbean genotypes viz., VC-3960 (A-88), VC-3960 (A-89), 98-CMH-016, NM-2 and BRM-195, and one blackgram genotype (VH-9440039-3) to be highly resistant to ULCD. A high phenol content and enhanced activity of polyphenol oxidase enzyme were shown to be associated with host plant resistance and are advocated as markers for ULCD resistance [74]. A blackgram variety VBN 8 released for cultivation in Tamil Nadu is claimed to be resistant to both crinkle as well yellow mosaic diseases [75]. A high yielding variety of urdbean-ADT6-A with moderate resistance to ULCD has been released for cultivation in rice fallows of the Cauvery delta region [76]. 

Researchers have shown that viral infections can lead to alterations in chlorophyll and sugar levels, resulting in a decline in photosynthetic activity [77,78]. Decreased photosynthesis has been associated with lower chlorophyll content in virus-infected leaves [79]. The viral infection can result in a decrease in chlorophyll content and an increase in sugar accumulation in both resistant and susceptible black gram cultivars. However, the effects can be more pronounced in susceptible cultivar compared to resistant cultivar [11]. Plants infected by the virus show increased soluble protein contents, which can be attributed to the activation of the host defense mechanisms and the attack mechanisms of the virus. The interaction between the virus and the plant triggers the production of proteins as part of the plant’s defense response. An increase in total soluble protein content upon ULCD infection was observed by few researchers [80,81]. However, protein content is known for lower accumulation in susceptible genotypes compared to resistant genotypes during viral infection [11,82,83]. Other studies concluded a decline in protein content in susceptible cultivars of the urdbean due to ULCD [84,85,86]. 

The first line of defense comprising lignin and suberin prevents the entry of invading pathogens by strengthening the cell walls of the host plant. Since both are derivatives of phenolic compounds, varied levels of these phenolic compounds can be observed in resistant and susceptible hosts [87,88]. It is known that phenols and their derivatives are harmful to plant pathogens. Their importance in pathogenic fungal, bacterial, and viral interactions in plants has been reported in earlier studies. These substances were reportedly found in greater concentrations in several wheat resistant germplasm and cultivars against fungal pathogens as a part of a defense mechanism [89]. In the case of ULCD infection, an increased concentration of phenolics in susceptible urdbean cultivars was observed, but they had insignificant role in resistance [90,91]. At an early-stage infection by ULCD in resistant cultivar, a reactive burst of ROS in the form of H_2_O_2_ was reported in abundance as compared to the susceptible cultivar [11]. A similar mechanism of the substantial burst of ROS is also known to be present in other pathosystems, including *Mungbean yellow mosaic India virus* (MYMIV) [92] and *Tobacco mosaic virus* (TMV) [93]. Enzymes such as ascorbate peroxidase (APX), peroxidase (POD), superoxide dismutase (SOD) and catalase (CAT) are synthesized in response to ROS to protect the host cells from oxidative stress [94,95]. The resistant urdbean cultivar was observed to upregulate SOD activity, leading to an accumulation of ROS at the time and site of infection, whereas the susceptible cultivar was observed to upregulate the CAT and APX activity leading to the prevention of ROS accumulation [11]. 

Another physiochemical change that generally occurs in plants on experiencing biotic stress as a part of a defense mechanism is the production of phytohormones such as salicylic acid (SA), jasmonic acid (JA) and ethylene (ET) [96,97]. ULCD-affected resistant cultivar was studied to have an upregulation in the expression of SA, JA and ET pathway-linked genes (*VmJAR1*, *VmAOC*, *VmWRKY*, *VmNPR1VmAP2/ERF* and *VmSAM*), exhibiting reduced symptoms as compared to the susceptible cultivar [11]. These linked genes act as a convergence point in the pathways of SA, JA and ET involved in defense mechanisms against viral infection, where ET plays a role in the modulation of either SA or JA [98]. Of these three phytohormones, the early activation of the SA pathway plays a major part in the activation of SAR (systemic acquired resistance) against viral pathogens [99]. A resistant cultivar may exhibit an increased expression of SA pathway-related genes such as *WRKY* and *NPR1* accompanied by a burst of H_2_O_2_ [100]. The SA induction may lead to higher activities of antioxidant enzymes (APX, SOD, and POD) [101], which were also observed by Karthikeyan et al., 2022 [11] against ULCD infection at an early stage in a resistant urdbean cultivar. JA pathway also acts as a key regulator of the defense mechanism against viral pathogens, but at the same time, it is antagonistic with the SA signaling pathway [102]. The late induction of JA pathway-related genes such as *JAR1* and *AOC* were observed in ULCD infection and concluded to be the responsible factor for the susceptibility reaction in the urdbean [11]. The time course of the expression level of genes associated with SA and JA pathways was observed to be different. Similar to the observations in TMV [103], SA pathway genes were activated at an early stage of ULCD infection, whereas those in the JA pathway were activated at a later stage and followed by the down regulation of early-activated SA pathway genes [11]. The resistance in ULCD-infected cultivars was further observed to be improved by the high expression of ET pathway genes (*AP2/ERF* and *SAM*) [11,104]. Collectively, the high expression of phytohormones pathway genes, reduction of ROS by antioxidants, and accumulation of soluble protein could be studied as key markers for understanding the resistance of the urdbean against ULCD. 

## 8. Causal Organism—Urdbean Leaf Crinkle Virus

From the early investigations ULCD is believed to be of virus etiology. Bhaktavatsalam et al. (1983) [105] in their investigation claimed that the causal agent of ULCD was neither a phytoplasma nor a viriod. Attempts have been made to characterize the causal virus(es) of ULCD. Sharma et al. (2014) [27] found an association of RNA containing flexuous rod-shaped virus particles measuring between 1600 nm and 2100 nm with a model length of 1950 nm. Whereas, Rao et al. (2002) [26] noticed spherical particles measuring *c*. 28 nm in diameter in the leaf dip preparations from the ULCD-affected leaves. They found that the antiserum of Blackgram mottle virus trapped a large number of particles in immunosorbent electron microscopy tests. Using ELISA, Patel et al. (1999) [106] found an indication of a causal agent of ULCD to be serologically related to the squash mosaic virus. Later, Baranwal et al. (2015) [107] employed next-generation sequencing (NGS) of the transcriptome from ULCD-affected urdbean plants. The sequence data revealed sequences of a virus origin that matched with *Cowpea mild mosaic virus* (*Carlavirus*), *Mungbean yellow mosaic India virus* (*Begomovirus*) and *Groundnut bud necrosis virus* (*Orthospovirus*). The latter two viruses are known to affect the urdbean and may be excluded as mere contaminants since the diseased samples were taken from the field. The role of the cowpea mild mosaic virus (CpMMV) in causing crinkle disease needs to be investigated using biological and molecular assays. The interference caused by asymptomatic viruses present as contaminants in field samples during the etiological identification of uncharacterized disease through NGS could be resolved by maintaining a pure inoculum under vector proof-controlled conditions. For this, in a recent study, an inoculation by the sprout seed abrasive method was developed and validated in different genotypes of the urdbean [108]. Earlier, the causal virus of ULCD was found to consist of spherical particles measuring 50 nm in diameter, which has similarities with *Pea enation mosaic virus* [109]. In ultra-structural studies of ULCD-affected leaves, virus-like particles (VLP), with a diameter from 25 to 30 nm, were noticed in the nucleus, cytoplasm and chloroplast [105]. Dubey et al. (1983) [62] purified a virus from the ULCD-affected urdbean and noticed spherical particles with an average diameter of 32 nm. The virus did not react with the antisera of 11 viruses belonging to the Bromo-, Poty- and Como virus groups infecting different pulse crops. 

## 9. Management Options

Attempts have been made to manage the ULCD using chemicals. Seed treatment with a combination of Imidacloprid, carboxin + tetramethylthiuram disulfide (TMTD), Pusa 5SD (a *Trichoderma virens*-based formulation), foliar sprays of imidacloprid and spinosad between 30 and 45 days after sowing (DAS), respectively, and a combination of seed treatment and foliar spray, all resulted in the reduced incidence of ULCD and increased grain yield in the urdbean [110]. Another study recommended a combined treatment strategy involving the heat therapy of seeds, as well as seed treatment with Imidacloprid 600 FS followed by spray with Imidacloprid 17.8 SL as an effective way of managing the leaf crinkle disease of the urdbean [111]. Plant extracts have also been used for the management of ULCD. Of the leaf extracts from 46 plant species, 10% leaf extracts of *M. jalapa* and *B. spectabilis* applied as a foliar spray before the inoculation of blackgram plants reduced ULCD. The reduction in disease was attributed to the induced resistance by the accumulation of phenols and enhanced activities of peroxidase, phenylalanine ammonia lyase and polyphenol oxidase in treated plants [112]. The plant extracts of *Zingeber officinalis*, *Prosopis juliflora* and *Piper longum* were found to inhibit an incidence of ULCD, and the inhibition was attributed to the induced resistance by the plant extracts [113]. An extract of ginger, bougainvillea and onion (at 2%) was also effective in reducing the incidence of ULCD, the former being the most effective followed by the latter [114]. In another study, neem (*Azadirachta indica*) and garlic (*Zingiber officinalis*) extract were found effective in reducing ULCD severity [115].

The pre-inoculation application of two strains of the plant-growth-promoting rhizobacterium (PGPR), *Pseudomonas fluorescens viz*., Pf1 and CHAO were found to be effective in inducing systemic resistance (ISR) against the causal agent of ULCD in blackgram (*Vigna mungo*), resulting in reduced ULCD infection [116]. Karthikeyan et al. (2009) [90] found that when six chemicals [benzoic acid, salicylic acid, acetyl salicylic acid, benzo (1,2,3) thiadiazole-7-carbothionic acid S-methyl ester (benzothiadiazole or BTH or Bion1) and thiamethoxam (Actara1)] and riboflavin at 100 ppm were sprayed on the 7-day-old plants 24 h before and after mechanical sap inoculation with a causal agent of ULCD and at the time of treatment (simultaneous application by mixing equal quantities of chemicals and infective sap) significantly reduced ULCD infection. The induced resistance was probably due to increased activities of peroxidase, phenylalanine ammonia lyase and polyphenol oxidase and the accumulation of phenolics in treated plants [90,112,116]. Nutrients play a vital role in the management of ULCD. It has been reported that plant defense mechanisms are regulated by growth regulators, and the contribution of nutrients such as N, P, K, Zn, B and Planofix (NAA) in the management of plant viruses, including the ULCD-causing agent, has been acknowledged by scientists. Therefore, balanced nutrient application can contribute to the management of ULCD and other plant viruses [91,117,118].

## 10. Conclusions

Viruses as disease-causing agents in plants have been known for more than a century. Since the discovery of the tobacco mosaic virus, which was the first plant virus to be characterized, about 1000 plant viruses have been identified [12,119]. Despite the large number of viruses known to infect different crops, outbreaks of new infectious viruses are still emerging, and newer viruses are still being characterized. It is imperative to know the identity of a virus before an effective preventive strategy is developed to ameliorate the losses caused by the outbreak of a new virus. Traditional methods of characterizing viruses [120] have been replaced by more sensitive, accurate and less time-consuming serological and PCR-based techniques for the identification of known and unknown plant viruses. These techniques, however, often fail to identify completely unknown virus(es). 

Of late, next generation sequencing (NGS) is being studied as an effective tool to decipher the identity of, not only an unknown virus, but of the whole virome of a plant in a short span of time [121,122,123]. The high cost involved, however, has restricted the use of this technology to a relatively small number of samples. The handling of a large amount of data generated also remains to be a challenge [121]. In fact, without the deployment of bioinformatics, the assembly of the complete genome from the large amount of data generated through next-generation sequencing of the virome is a cumbersome task. NGS data with the help of bioinformatics can be utilized to decipher the identity of the virus or viruses associated with ULCD in different parts of the country. Once the identity of the virus/es is ascertained, wet lab studies can unequivocally settle the role of the virus/es in causing ULCD. There are reports of aphids and whiteflies transmitting the urdbean leaf crinkle virus in different parts of India. Further, there are publications on the morphology of the virus particles wherein spherical [26,62,105] and flexuous rods [27] have been shown to be associated with the crinkle-affected urdbean. Thus, the available information on vectors of ULCD and the morphology of the causal virus points to the possibility of the involvement of different viruses in causing the crinkle disease of urdbeans. Thus, the virome of ULCD-affected urdbean plants from different geographical areas needs to be studied in order to solve the mystery behind the virus/es responsible for causing ULCD in the urdbean. 

## Figures and Tables

**Figure 1 viruses-15-02120-f001:**
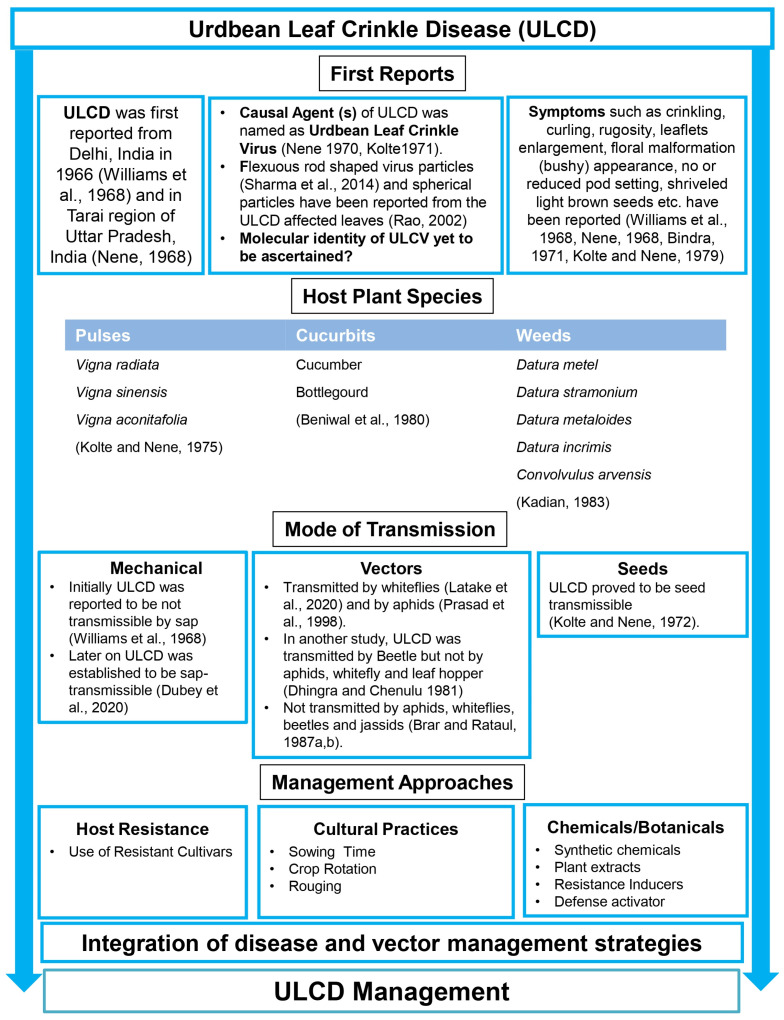
Schematic presentations of the various aspects of urdbean leaf crinkle disease. References: Beniwal et al., 1980 [13]; Bindra 1971 [14]; Brar and Rataul, 1987a, b [15,16]; Dhingra and Chenulu, 1981 [17]; Dubey et al., 2020 [18]; Kadian 1983 [19]; Kolte 1971 [20]; Kolte and Nene, 1972, 1975, 1979 [21,22,23]; Latake et al., 2020 [24]; Nene 1968, 1970 [8,10]; Prasad et al., 1998 [25]; Rao 2002 [26]; Sharma et al., 2014 [27]; and Williams et al., 1968 [7].

**Figure 2 viruses-15-02120-f002:**
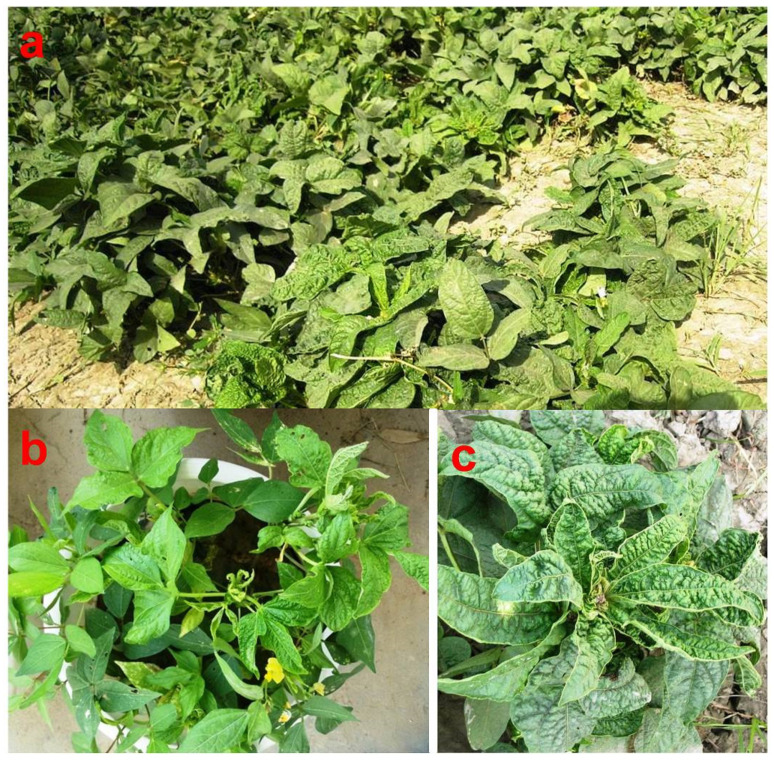
Urdbean leaf crinkle-affected plants in field (**a**), symptoms in mechanically sap-inoculated urdbean plant (**b**) and close up of an affected plant showing rugosity in leaves (**c**).

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
