# Peer review of "Urdbean Leaf Crinkle Virus: A Mystery Waiting to Be Solved"

_viruses, 2023, doi:10.3390/v15102120_

Round 1
Reviewer 1 Report
viruses-2634598
Review article
Title: Urdbean leaf crinkle disease: A mystery waiting to be solved
This document offers a thorough examination of urdbean leaf crinkle disease (ULCD), with the goal of compiling and analyzing the limited information available on its symptoms, epidemiology, host range, host physiochemical changes, and possible causes. The review is well-organized and easy to understand, with a sufficient collection of cited literature that covers research on ULCD. The topic is timely and pertinent.
As a reviewer, I have noticed that sections 7 and 8 of the manuscript are the least accomplished parts. The information presented in these sections appears to be separated without any evident reason. Section 7 starts by discussing general effects and responses to plant viral diseases and possible resistance mechanisms. However, it then moves on to present a long list of genotypes and their putative resistance reactions, without providing any evidence related to the initial paragraphs. In section 8, while reviewing virus-induced alterations in host physiochemical parameters, the topic of resistance is revisited, and some evidence related to ULCD is discussed. I believe that combining the information from sections 7 and 8 into one section would greatly benefit the manuscript, making it more cohesive and better articulated.
Reviewer 2 Report
The manuscript “Urdbean leaf crinkle disease: A mystery waiting to be solved “by Naimuddin and colleagues summarized the information on the various aspects of ULCD. There are several problems including but not limited those suggestions listed below, that will require the authors attention and revise before publishing in Viruses.
1. Line 14: without conforming to Koch’s postulates- without conforming by/though Koch’s postulates
2. Line 14: delete still, it is redundant with remains
3. Line 24 25: “kharif” and “rabi” season, could be change to an easier understanding and wildly used word?
4. Figure 1: the colors are too cluttered, please revise to be succinct and clear
5. Line 111: summer sown crops “and”?
6. Line 123: what is “contradictions vis~a~vis mechanical”?
7. Line 148 and 151: Per cent-Percent
8. Line 155: without any incubation? Unclear description
9. Line 161: “Another aphid, Myzus persicae was also shown to transmit ULCD” can be integrated into line 158
10. Line 188-196: Most commonly found possible secondary hosts of ULCD such as including……have been studied to exhibit resistance against the causal agent of ULCD? Secondary host exhibit resistance against ULCD, so can be regard as host? ULCD can affect and show symptoms in those secondary host?
11. Line 297: to be easy read and understanding, “Causal organism” part should be placed in the front of MS, although the really causal agent of ULCD is inconclusive
12. Line 329: it has two “,”
13. Line 353: 100ppm-100 ppm
14. Line 353: what is the mean of “@”?
15. Line 373: these methods indicate tranditional methods or serological and nucleic acid based techniques ? it has an ambiguity
16. Line 379” two “.”
Minor editing of English language required
